# Compound and Conditioned Likelihood Ratio Behavior within a Probabilistic Genotyping Context

**DOI:** 10.3390/genes13112031

**Published:** 2022-11-04

**Authors:** Kyle Duke, Daniela Cuenca, Steven Myers, Jeanette Wallin

**Affiliations:** California Department of Justice, Richmond, CA 94804, USA

**Keywords:** likelihood ratio, sub-source propositions, conditioning, compound LR, additive behavior

## Abstract

In cases where multiple questioned individuals are separately supported as contributors to a mixed DNA profile, guidance documents recommend performing a comparison to see if there is support for their joint contribution. Anecdotal observations suggest the summed log of the individual likelihood ratios (LR), termed the simple LR product, should be roughly equivalent to or less than the log(LR) for the joint likelihood ratio, termed the compound LR. To assist casework analysts in evaluating statistical weights applied to a case at hand, this study assessed how consistently compound LRs conform to an additive behavior when compared to the simple LR product counterparts. Two-, three-, and four-person DNA mixture data, of various mixture proportions and DNA inputs, were interpreted by STRmix^®^ version 2.8 Probabilistic Genotyping Software. Relative magnitudes of LR increases were found to be dependent on both template level and mixture composition. The distribution of log(LR) differences between all compound/simple LR comparisons was ~−2.7 to ~28.3. This level of information gain was similar to that for compound LR comparisons, with and without interpretation conditioning (~−3.2 to ~27.7). In both scenarios, the probability density peaked at approximately 0.5, indicating the information gain from constrained genotype combinations has a comparable impact on the outcome of LR calculations whether the restriction is applied before or after interpretation.

## 1. Introduction

Likelihood ratios (LRs) have been in use in forensic DNA typing for several decades now as a weight-of-evidence statistic for DNA mixture interpretation [1,2,3,4,5,6,7]. The construction of a sub-source LR necessarily requires a consideration of which propositions best represent the inclusionary (prosecution) and alternative (exclusionary/defense) theories about the source of the DNA evidence based on the relevant case information [8,9,10,11]. When the inclusionary proposition includes one individual as a contributor to the mixture and the alternative proposition replaces that individual with an unknown, unrelated contributor, the LR propositions for such a calculation, hereafter referred to as a “simple LR”, can be represented with the following shorthand notation (shown below for a two-person mixture), where “ID” represents a questioned individual and “U” represents an unknown, unrelated mixture contributor:(1)$$\frac{\mathrm{Inclusionary}\;\mathrm{proposition}}{\mathrm{Alternative}\;\mathrm{proposition}}=\frac{\mathrm{ID}1+U}{U+U}\;$$

This LR assesses how well individual ID1 explains the data.

When there are multiple individuals in question and the presence of at least one of those individuals in the evidence can be reasonably postulated, an LR for the remaining individuals, hereafter referred to as a “conditioned LR”, may be calculated [12,13,14,15]. Basic properties of conditioned LRs in the forensic DNA context have been previously discussed in the literature. Generally, when the simple LRs for multiple individuals are greater than 1, conditioning on one or more individuals increases the magnitude of the LR for the remaining individuals, if all individuals are true contributors to the mixture [16]. The higher value from a conditioned LR is due to the reduction in ambiguity that results from a decrease in the possible genotype combinations in the mixture. This basic concept is represented below as: Unconditioned LR for ID1 LR for ID1, Conditioned on ID2
(2)$$\;\frac{\mathrm{ID}1+U}{U+U}\;\;\leq \;\;\frac{\mathrm{ID}1+\mathrm{ID}2}{U+\mathrm{ID}2}$$
where ID1 and ID2 represent two different individuals.

On the other hand, the most appropriate proposition pair in the context of a given case may place all questioned individuals in the inclusionary proposition versus an equal number of unknown, unrelated contributors in the alternative proposition. The propositions for this calculation, hereafter referred to as a “compound LR”, are represented below (with the same two-person mixture as an example):(3)$$\frac{\mathrm{Inclusionary}\;\mathrm{proposition}}{\mathrm{Alternative}\;\mathrm{proposition}}=\frac{\mathrm{ID}1+\mathrm{ID}2}{U+U}\;$$

A criticism by the recent NISTIR 8351-DRAFT [17] speaks to the impact of specific propositions chosen for the calculated LR value: “This fact should encourage more effort to standardize development of propositions…” Numerous publications on proposition formulation and LR reporting [10,11,12,15,18] provide valuable guidance. While simple, conditioned, and compound likelihood ratios are described in detail, the relative quantitative properties are not well represented in the literature. In particular, it has anecdotally been observed that when simple LRs (>1) for different individuals in a given case are multiplied together (i.e., their exponents are added), the product, hereafter referred to as the “simple LR product”, is typically less than or equal to the corresponding compound LR for the combination of those same individuals. This additive behavior is represented below, using the same two-person mixture from Equations (1) and (3): Simple LR product for ID1 and ID2   Compound LR for ID1 and ID2
(4)$$\;\;\;\;\frac{\mathrm{ID}1+U}{U+U}*\frac{\mathrm{ID}2+U}{U+U}\;\;\;\leq \;\;\;\frac{\mathrm{ID}1+\mathrm{ID}2}{U+U}$$

A related question is whether a compound LR increases when adding a conditioned contributor, as represented below with a three-person mixture:(5)$$\frac{\mathrm{ID}1+\mathrm{ID}2+U}{U+U+U}\;\leq \;\frac{\mathrm{ID}1+\mathrm{ID}2+\mathrm{ID}3}{U+U+\mathrm{ID}3}$$

Exploring the robustness of these quantitative relationships is helpful both to individual casework analysts making decisions about setting propositions for compound LRs as well as standardization bodies that require empirical support for guidelines related to these LRs.

Presented here is a systematic assessment of compound and conditioned LR calculations aimed at establishing LR magnitude expectations. One of the primary objectives of this study was to determine how consistently compound and conditioned LRs conformed to an additive behavior across a variety of mixture proportions as well as a wide range of DNA inputs. Beyond the question of whether a compound or conditioned LR is equal to or greater in magnitude than its simple LR product/unconditioned counterpart, the study sought also to address the questions of how large the increase is and whether the relative magnitude of the increase can be predicted based on the properties of the mixture. Finally, the study assessed whether any parallels can be drawn between the magnitudes of compound and conditioned LRs. Such information can assist the casework analyst in evaluating the statistical weights of the LRs calculated in a case at hand. For example, in sexual assault or touch DNA cases where the prosecution’s case posits multiple suspects as DNA mixture contributors, empirically supported expectations of compound and/or conditioned LR magnitude would allow an analyst to confidently provide information on whether those multiple suspects can explain the mixture data when considered together, instead of considering them in isolation. Per the authors of the NISTIR 8351-DRAFT, such information is currently insufficient in the literature.

## 2. Materials and Methods

### 2.1. Construction, Amplification, Capillary Electrophoresis, and Analysis of Ground Truth DNA Mixtures

Buccal cell DNA was collected with informed consent from healthy, unrelated laboratory volunteers and extracted using the PrepFiler™ DNA Extraction Kit (Life Technologies, South San Francisco, CA, USA).

Because two-person mixtures are the easiest for STRmix^®^ to analyze, the composition of the two-person donor sets was not directed at achieving a given maximum allele count (see Table 1 and Appendix A). Meanwhile, the three- and four-person mixture donors were selected so that one set that had a maximum allele count consistent with N-1 contributors-, i.e., a maximum of four detected alleles per locus for the three-person mixtures and a maximum of six detected alleles per locus for the four-person mixtures-and one set that had a maximum allele count consistent with N contributors (5–6 alleles per locus for the three-person mixtures and 7-8 alleles per locus for the four-person mixtures).

Mixtures at a variety of DNA input amounts and mixture ratios were constructed by quantifying extraction yield for each donor with Quantifiler™ Trio DNA Quantification Kit (Life Technologies) and combining the amounts of template indicated in Table 2. Each template series for a given mixture was made by diluting a single high-level mixture. Particular attention was focused on the inclusion of mixtures with high major to minor contributor ratios (e.g., 99:1, 100:100:4, 100:100:100:6), since these extreme ratios are occasionally encountered in casework and the PCAST review of forensic science [19,20] indicated that more empirical support is needed to demonstrate reliable software performance under these conditions: “… the studies involve few mixtures in which a sample is present at an extremely low ratio. By expanding these empirical studies, it should be possible to test validity and reliability across a broader range”.

The mixtures were amplified with the GlobalFiler™ PCR Amplification Kit (Life Technologies), either singly or in duplicate as indicated, with a 28-cycle thermal cycling protocol on the ProFlex (Life Technologies) and electrophoresed on an Applied Biosystems™ 3500 Genetic Analyzer using POP-6 polymer (Life Technologies). Replicate amplifications were performed for any mixture with a minimum minor mixture proportion less than 25% in order to test a feature of STRmix unrelated to this study; however, since these replicates are independent data points, they were all included in this study, and each was independently interpreted with STRmix.

The capillary electrophoretic data from the amplified mixtures were subsequently analyzed in GeneMapper ID-X v1.6 (Life Technologies, South San Francisco, CA, USA) at channel-specific analytical thresholds of 51 RFU (blue), 71 RFU (green), 35 RFU (yellow), 41 RFU (red), and 61 RFU (purple) and analyzed with STRmix v2.8 (Institute of Environmental Science and Research, Auckland, New Zealand) using laboratory-validated settings, which include stutter models for −1/+1, −2/+2, −0.5/+0.5, and −1.5 STR repeats. Upon completion of interpretation, the targeted mixture proportions were confirmed to be consistent with the mixture proportions from the STRmix Interpretation Report for the highest template amount for each mixture from Table 2 (data not shown).

For this study, both unconditioned and conditioned analyses were performed, with the conditioned contributors consisting of each combination of single and multiple mixture donors up to a total number of N-1. The “MCMC Accepts” settings used were the default MCMC accepts of 10,000 burn-in/50,000 post burn-in per chain unless stated otherwise, where the accepts were increased by a factor of 20 to 200,000 burn-in and 1,000,000 post burn-in per chain.

### 2.2. Compound LR Calculations

Compound LR calculations were performed for all combinations of true contributors to the 197 mixtures in Table 2 using STRmix v2.8 [21]; this amounts to a total of 1775 compound LRs calculated (see Appendix A for raw LR data). Table 3 is a summary of the proposition sets used for these calculations. Note that all combinations of two contributors were tested for the three-person mixtures, and all combinations of two and three contributors were tested for the four-person mixtures. One-sided 99.9% lower credible intervals, termed “Highest Posterior Density” (HPD), were calculated for all LRs unless otherwise indicated. NIST databases for African American, Caucasian, and Hispanic populations [22] were used for the calculations. Equal proportion population-stratified LRs were reported as the primary weight-of-evidence statistics.

Simple LRs for all contributors to the mixtures in Table 2 were calculated, and the log(LR) were plotted to compare the magnitude of each compound LR to its corresponding simple LR product for each contributor combination.

### 2.3. Conditioned LR Calculations

LR calculations were performed using STRmix v2.8 for all combinations of non-conditioned true contributors to the 197 mixtures in Table 2, given a particular conditioned interpretation; this amounts to a total of 4686 conditioned LRs calculated (see Appendix A for raw LR data). Table 4 summarizes the proposition sets used for these calculations.

Corresponding unconditioned LR calculations were performed, and the log(LR) were plotted comparing the magnitude of each conditioned LR to its corresponding unconditioned LR for each contributor combination.

## 3. Results

### 3.1. Compound LR Calculations: 2- and 3-Person Mixtures

The results of the compound LR calculations show that the additive behavior of compound LRs largely holds over the entire per-contributor DNA input range of ~5 pg–2.5 ng for 2- and 3-person mixtures (see Figure 1), with a majority above the black line of equivalence (~92% and ~88%, respectively). Furthermore, the majority (100% for 2-person and 94% for 3-person) of the compound LRs that are less than their simple LR product counterparts, were within one order of magnitude from the line of equivalence.

In terms of the trends observed for the various mixture proportions, the compound LRs with the largest difference are those with the most ambiguity in the genotype combinations (i.e., the mixtures with the most even mixture proportions). The 2-person mixtures are so well resolved that only the higher level 1:1 mixtures, with total DNA input levels of 100 pg, 200 pg, 400 pg and 800 pg, show a substantial increase in the compound LR compared to the simple LR product, whereas the 3-person compound LRs show significant increase over the simple LR product for most mixture proportions.

### 3.2. Conditioned LR Calculations: 2- and 3-Person Mixtures

LRs for true contributors to the 2- and 3-person mixtures interpreted with conditioning were generally equal to or higher than their unconditioned counterparts (see Figure 2). This effect is expected due to the simplification imposed upon the interpretation by fixing one or more contributor genotypes. Comparing these plots to the compound LR plots, the same trends in terms of the mixture proportions most affected by conditioning are apparent; the mixtures with the most ambiguity in the genotype combinations (i.e., the mixtures with the most even mixture proportions) tend to increase the most with conditioning.

### 3.3. Compound LR Calculations: 4-Person Mixtures

The compound LR plots for the 4-person mixtures also show that the additive behavior generally holds over the entire per-contributor DNA input range of ~5 pg–2 ng, as well as the same trends in upward deviation from the line of equivalence with increasing ambiguity of genotype combinations. However, many false negative compound LRs (defined as LRs of 0) are also apparent, which increase in frequency with the number of true contributors placed in the numerator of the LR (see Figure 3).

A critical observation in understanding the root cause of these false negative compound LRs is where they fall on the x axis compared to the rest of the data set. The false negatives occur at relatively low log(simple LR product) values, but do not primarily occur near the origin. Because LR magnitude correlates with DNA mixture input, this simple LR product range corresponds to mixtures that are relatively low level but not the lowest level in the data set. For instance, among the Donor Group 1 4-person mixtures, the false negatives in Figure 3c had modest total DNA input ranges of ~250–638 pg (~6–250 pg on a per-contributor basis); the Donor Group 2 false negatives had a similar total DNA input range of ~200–625 pg (~5–350 pg on a per-contributor basis). In terms of the complexity of the interpretation (and the corresponding length of the plausible genotype combination list generated by STRmix), these mixtures ranked among the most complicated, because the contributors’ input amounts are all in the stochastic range-high enough to allow full allelic detection of some contributors but low enough that allelic dropout is a realistic (and in some cases expected) possibility for all contributors.

Concomitant with the false negative compound LRs for the 4-person mixtures was an effect on the magnitude of the difference between the point estimate LR and HPD LR, which can be viewed as a secondary diagnostic of the issue. The 2- and 3-person plots of HPD LR v. point estimate LR (see Appendix B Figure A1) were as expected, given that the HPD is intended to be a lower bound on normal LR variation; a log-scale drop of 1–2 units from point estimate to HPD was observed for the compound LRs, the same as observed for the simple LRs also shown in the plots. However, very large differences between the point estimate LR and HPD LR (>5 log units, in some cases >10 log units) were observed in the 4-person HPD v. point estimate plots of the compound LRs, with the outlying data points increasing along with the number of contributors in the LR numerator (Appendix B Figure A2a).

The increased complexity of the problematic 4-person mixture data, relative to the remaining 4-person data, explains both where the false negative compound LRs emerge in terms of x axis position and frequency, as well as why they are accompanied by large differences between the point estimate and HPD LRs. With a very complex interpretation and a long list of plausible genotype combinations comes a substantial diffusion of probability among the options. Because obtaining a non-zero LR requires that the interpretation places all of the true contributors together in the mixture, with a longer list of plausible options comes an increasingly remote opportunity for the true genotype combinations to be selected at all loci over the course of the interpretation. Hence, the sample effectively hits a “complexity ceiling”. In other cases, even if a non-zero LR is avoided, the same diffusion of probability leads to the very large drop in HPD, as very small genotype combination weights can lead to excessively low resampled weights from the HPD calculation process.

### 3.4. Optimization of Markov Chain Monte Carlo (MCMC) Accept Settings for 4-Person Compound LRs

If the complexity ceiling is the issue causing the false negative LRs and large differences between the point estimate LR and HPD LR, then more thoroughly exploring the probability space with increased MCMC accepts is a reasonable solution that has been suggested in the relevant literature [23,24]. In order to preliminarily test a MCMC accept increase, we reanalyzed all of the problematic 4-person mixtures with the MCMC burn-in and post burn-in accepts set to 100,000 and 500,000 per chain, which is a 10-fold increase over the default settings of 10,000 burn-in and 50,000 post burn-in accepts per chain for STRmix v2.8. These altered settings produced non-zero LRs for every problematic mixture with the exception of a 250 pg 4:3:2:1 mixture. In this case, the LR with all four contributors in the LR numerator was 0, as was the LR with the three lower-level contributors in the numerator (at input levels of 75 pg, 50 pg, and 25 pg), indicating that fitting these low-level contributors together was the source of the issue. Upon performing 10 replicate analyses and 4-person compound LR calculations for this mixture, 3 of the 10 recalculations returned LRs of 0, indicating that further adjustment of the MCMC accepts was warranted.

To optimize the MCMC accepts to completely eliminate the observation of false negative compound LRs, the variations in burn-in and post burn-in accepts shown in Table 5 were tested on the 250 pg 4:3:2:1 sample. We observed that while increasing the post burn-in accepts 10-fold alone was more effective in preventing false negative LRs than increasing the burn-in accepts 10-fold alone, false negatives were still observed after both isolated setting adjustments. There was at least one false negative compound LR with a 20x increase in burn-in or post burn-in, when adjusted in isolation. However, no false negative compound LRs were observed with a 15x, 20x or 50x increase to both burn-in and post burn-in accepts. In order to achieve a balance between eliminating false negative compound LRs and avoiding unnecessary increases in runtime, we elected to move forward with the 20x MCMC accept increase (i.e., to 200,000 burn-in/1 million post burn-in per chain) for the remainder of the study.

### 3.5. Repeated Interpretation/Compound LR Calculation at 20x MCMC Accepts

All of the 4-person interpretations and calculations from Figure 3 were performed with the adjusted 20x MCMC accepts setting in place (see Figure 4). No LRs of 0 were observed, and although some large differences between point estimate and HPD LR were still observed, their incidence and magnitude were greatly reduced (see Appendix B Figure A2).

### 3.6. Conditioned LR Calculations: 4-Person Mixtures

Observations of false negative LRs occurred for the conditioned 4-person interpretations with the default number of MCMC accepts, similarly to the compound 4-person LRs (see Figure 5). A prominent difference with the conditioned 4-person LR plot is that the false negatives occurred along both axes; that is, in some instances, unconditioned compound LRs were zero and the corresponding conditioned LRs were non-zero, whereas in other cases unconditioned compound LRs were non-zero and the corresponding conditioned LRs were zero.

Another difference in the conditioned 4-person LR plot is the reversal of the trend observed for increasing numbers of contributors in the LR numerator; whereas false negatives increased along with the number of contributors in the numerator, false negatives decreased with increased amounts of conditioning. This effect would be expected, given that increased conditioning correlates with decreased mixture complexity.

### 3.7. Conditioned LR Calculations: 4-Person Mixtures at 20x Increased MCMC Accepts

The interpretations and LR calculations from Figure 5 were performed again at 20x MCMC accepts. As with the compound LR plots, the increased accepts effectively addressed the false negative LRs observed with the default accepts; however, they were not completely eliminated (see Figure 6). Examining the EPG data for the conditioned interpretations with false negative LRs at 20x MCMC accepts (see Appendix B Figure A3) demonstrates a biological cause for these false negatives. In both cases, a very low level contributor produced unexpectedly high peak data at one locus, given the targeted mixture proportions, and the high peak data was produced for an allele *exclusive to* that low level contributor.

Under these amplification conditions, STRmix heavily favored sharing the low level contributor peak. Further exacerbating the drive toward an incorrect inference was the fact that in both cases, the low level contributor in question was conditioned upon. Since data at other loci were consistent with an overall low contribution to the mixture, sharing of the peak appeared to be the most consistent with the data, despite this not being the ground truth.

Since the reason for the false negative LRs from the conditioned interpretations is an aberrant PCR amplification, the issue would be unlikely to be effectively addressed by increasing MCMC accepts. In other words, it is the unexpected height of the low level contributor peaks, not the length of the genotype combination list that is causing genotype combinations inconsistent with ground truth to be favored by the software.

## 4. Discussion

The data indicate that compound and/or conditioned LRs can be expected to be greater than simple LR products in a vast majority of cases, whether the mixture in question is at a low or high level of DNA input. Furthermore, the typical amount of increase was dependent on both the DNA input level as well as the mixture composition.

The spread of the data around the equivalence line in each plot is small near the origin, then expands in the middle of the x-axis range before contracting again near the upper boundary of the x-axis range. This indicates that maximum information gain from compound LR calculation and/or conditioning occurred at input amounts in the middle of the studied input range, which on a per-contributor basis is ~1 ng for our dataset. Such a maximum is sensible, given that low-level mixtures contain less information in general, and high-level mixtures are more definitively analyzed and so have less information to gain from compound LR calculation or conditioning than mixtures at a moderate template levels.

The impact of mixture composition on the magnitude of LR change for both the compound and conditioned LRs is most apparent when the constraints placed on the genotype combinations relevant to the LR calculations are maximized; that is, when N true contributors are placed in the numerator of an N-contributor mixture and when N-1 contributors are conditioned upon in the interpretation of an N-contributor mixture. Figure 1a,c and Figure 2a,c, for the 2- and 3-person mixtures, are the most striking in this regard; mixtures with contributors in the same proportions have regular, distinct datapoint groupings, and the trends for the mixtures with more inherent ambiguity (e.g., 1:1, 1:1:1 and 100:100:4) are translated further upward than the trends for the more easily resolved mixture proportions. The corresponding Figure 4c and Figure 6c, for the 4-person mixtures, have less clear demarcation of the datapoint groupings for the ambiguous mixture proportions, which is not surprising given their level of complexity.

Because not every compound LR was higher than its simple LR counterpart, our expectations about compound LR magnitude are most straightforwardly defined in terms of upper and lower bounds. The distribution of log(LR) differences between all compound LRs in this study and their corresponding simple LR products ranged from a minimum of ~−2.7 to a maximum of ~28.3, with the probability density peaking at approximately 0.5 (see Figure 7a).

Furthermore, in a similar fashion to the compound/LR product plots, the conditioned/unconditioned LR plots show that conditioning does not always lead to a higher LR. The distribution of log(LR) differences between conditioned LRs and their corresponding unconditioned counterparts ranged from a minimum of ~−3.2 to a maximum of ~27.7, with the probability density again peaking at approximately 0.5 (see Figure 7b).

The similarity of the log(LR) difference distributions for the compound/simple and conditioned/unconditioned comparisons (see Figure 7c) points to parallel mechanisms of action. In fact, because the weights produced by STRmix for the various genotype combinations can be characterized as either unconditional or conditional probabilities, some connections can be drawn between the effects of compound LRs and conditioned LRs.

For instance, if we let Pr(A) be the weight assigned to a genotype combination including contributor A, we could also let Pr(A|B) be the weight assigned to the same genotype combination if the interpretation were conditioned on contributor B. Alternatively, if we place both contributor A and contributor B in the numerator of the LR for a compound LR calculation, we could let Pr(A,B) be the weight assigned to the combination of contributor A and contributor B together. However, we could also deconstruct Pr(A,B) into Pr(B) * Pr(A|B).

Since the same type of conditional probabilities come into play in calculating both compound LRs and LRs based on conditioned interpretations, this explains why the information gain from performing these types of calculations is similar. In fact, it could be said that calculating a compound LR is performing a type of conditioning, the difference being that the conditioning is applied at the point of LR calculation rather than at the point of interpretation.

## 5. Conclusions

The relative quantitative results of this LR study provide a basis for expecting additive behavior of LR exponents at all DNA input amounts. While not all of the compound LRs were greater than their simple LR product equivalents, a vast majority were equal or greater, and trends in the magnitude of the effect were seen in relation to both template DNA amount and mixture composition. We also determined that the substantial increases in analysis complexity inherent to higher-order mixture interpretations may necessitate increased MCMC accepts in order to avoid false negative compound LRs. In addition, the similarity of the log(LR) distributions for compound LRs and conditioned LRs relative to their simple LR product and unconditioned counterparts, respectively, derives from the similar information gain that occurs from either conditioning during the interpretation or the LR calculation of a compound LR.

## Figures and Tables

**Figure 1 genes-13-02031-f001:**
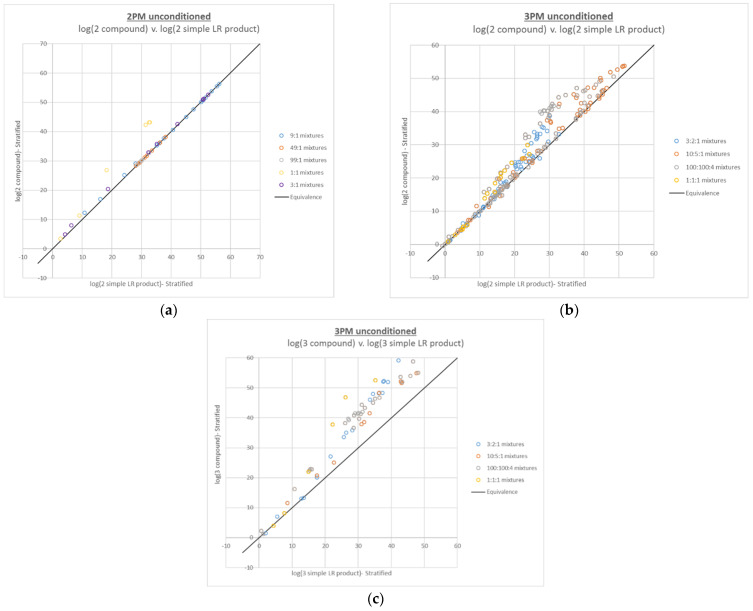
Plots of log(compound LR) v. corresponding log(simple LR product) for (**a**) 2-person mixtures with 2 true contributors, (**b**) 3-person mixtures with 2 true contributors, and (**c**) 3-person mixtures with 3 true contributors.

**Figure 2 genes-13-02031-f002:**
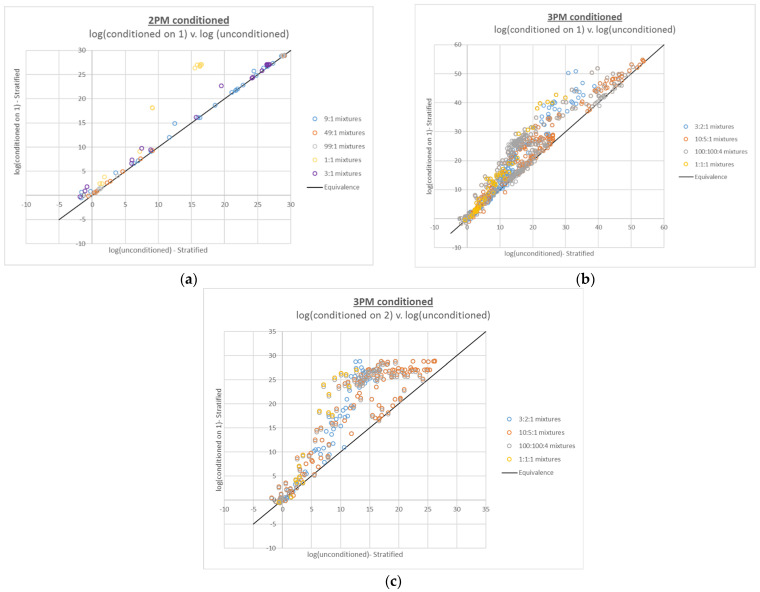
Plots of 2- and 3-person log(LR)s based on conditioned interpretations v. corresponding log(LR)s based on unconditioned interpretations: (**a**) 2-person mixtures conditioned on 1 true contributor, (**b**) 3-person mixtures conditioned on 1 true contributor, and (**c**) 3-person mixtures conditioned on 2 true contributors.

**Figure 3 genes-13-02031-f003:**
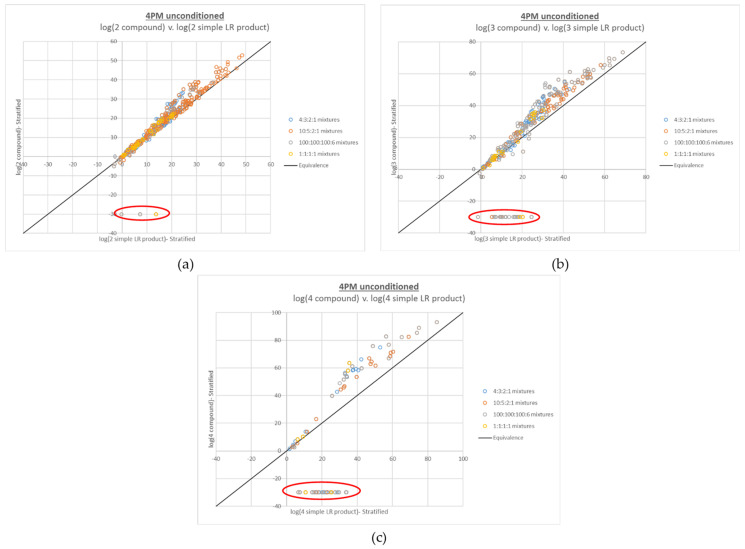
Plots of log(compound LR) v. corresponding log(simple LR product) for 4-person mixtures, with (**a**) 2 true contributors, (**b**) 3 true contributors, and (**c**) 4 true contributors in the numerator proposition. Increasing numbers of false negative compound LRs (LRs of 0, plotted at −30 and circled in red) were observed with increasing numbers of true contributors in the LR numerator. Approximately 0.7% of the data points in (**a**), 7.4% of the data points in (**b**), and 31% of the data points in (**c**) are false negative LRs.

**Figure 4 genes-13-02031-f004:**
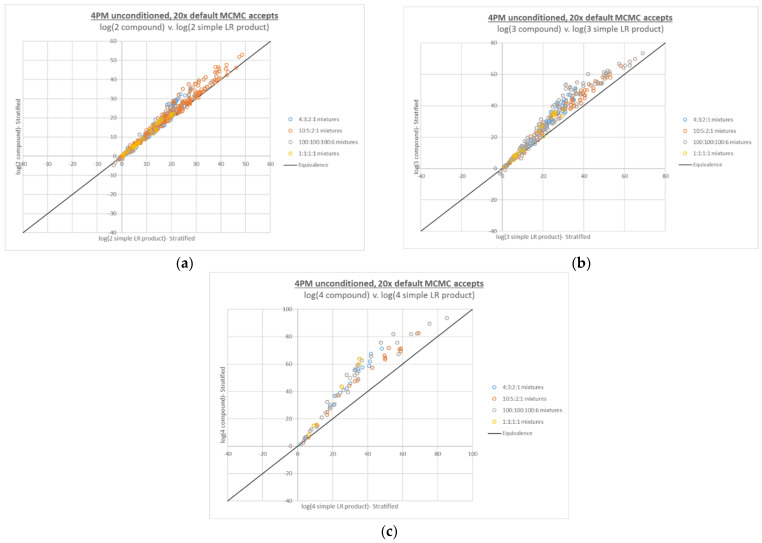
Four-person compound LR plots from Figure 3, recalculated based on interpretations run at 20x MCMC accepts, with: (**a**) 2 true contributors, (**b**) 3 true contributors, and (**c**) 4 true contributors in the numerator propositions.

**Figure 5 genes-13-02031-f005:**
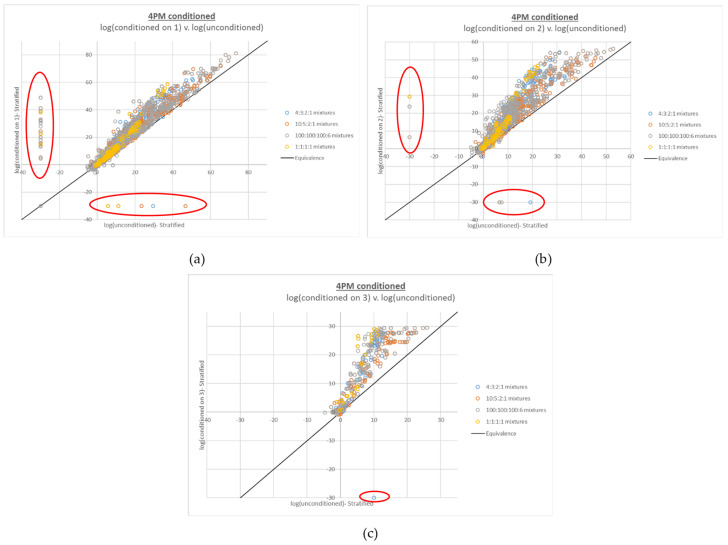
Plots of 4-person log(LR)s based on conditioned interpretations v. corresponding log(LR)s based on unconditioned interpretations run at the default number of MCMC accepts. The number of conditioned true contributors are (**a**) 1, (**b**) 2, and (**c**) 3. False negative LRs (LRs of 0, plotted at −30 and circled in red) are observed along both the X and Y axes for all but the interpretations conditioned on three true contributors (**c**).

**Figure 6 genes-13-02031-f006:**
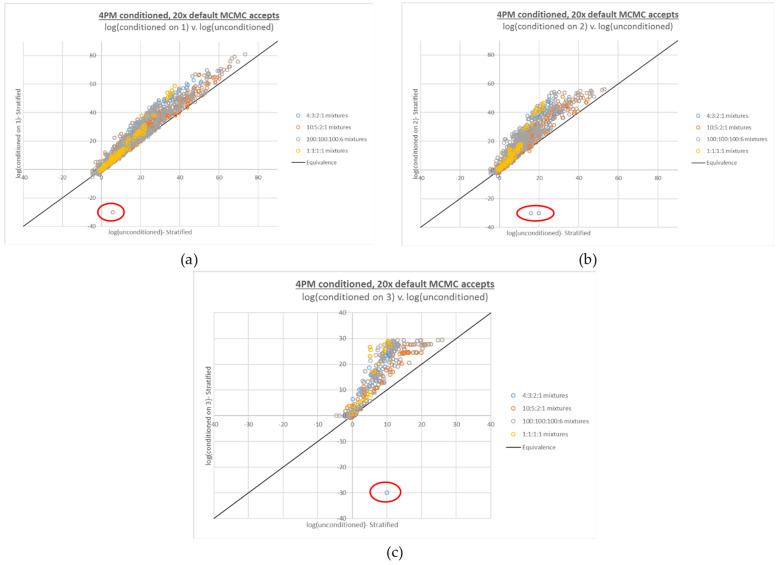
Four-person conditioned LR plots from Figure 5, recalculated based on interpretations run at 20x MCMC accepts. The number of conditioned true contributors are (**a**) 1, (**b**) 2, and (**c**) 3. A limited number of false negative LRs (LRs of 0, plotted at −30 and circled in red) are still observed with 20x accepts.

**Figure 7 genes-13-02031-f007:**
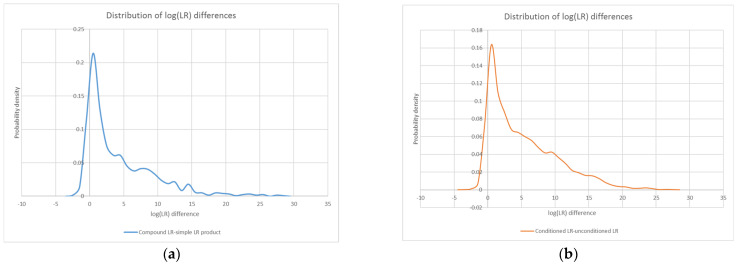
Probability density distributions for the log difference between all compound LRs in the study and their corresponding simple LR products (**a**), as well as all conditioned LRs in the study and their unconditioned counterparts (**b**). An overlay of the two distributions (**c**) highlights their similarity.

**Table 1 genes-13-02031-t001:** DNA donors for compound/conditioned LR study mixtures.

Donor Group (Donor Number)	Max Number of Alleles	Donor ID	Sex
2-person Group 1 (1)	4	F1	Female
2-person Group 1 (2)	4	F2	Female
2-person Group 2 (1)	4	F3	Female
2-person Group 2 (2)	4	M1	Male
3-person Group 1 (1)	4	M2	Male
3-person Group 1 (2)	4	F1	Female
3-person Group 1 (3)	4	F2	Female
3-person Group 2 (1)	6	F4	Female
3-person Group 2 (2)	6	F3	Female
3-person Group 2 (3)	6	M1	Male
4-Person Group 1 (1)	6	M3	Male
4-Person Group 1 (2)	6	F5	Female
4-Person Group 1 (3)	6	M4	Male
4-Person Group 1 (4)	6	F6	Female
4-person Group 2 (1)	8	M5	Male
4-person Group 2 (2)	8	F7	Female
4-person Group 2 (3)	8	F8	Female
4-person Group 2 (4)	8	F9	Female

**Table 2 genes-13-02031-t002:** Sample composition of mixtures for compound/conditioned LRs. Donor mixture ratios are listed from left to right in accordance with the donor numbers listed in Table 1. The input amounts listed are for total DNA.

Donor Group	Mixture Ratio	Input Amounts Tested	Replicates
2-person Group 1	9:1	2 ng, 1 ng, 870 pg, 750 pg, 500 pg, 380 pg, 250 pg, 125 pg, 63 pg	2
2-person Group 1	49:1	2.5 ng, 1.9 ng, 1.25 ng, 625 pg, 313 pg	2
2-person Group 1	99:1	2.5 ng, 1.25 ng, 625 pg	2
2-person Group 2	1:1	800 pg, 400 pg, 200 pg, 100 pg, 50 pg, 25 pg	1
2-person Group 2	3:1	800 pg, 400 pg, 348 pg, 300 pg, 200 pg, 152 pg, 100 pg, 50 pg, 25 pg	1
3-person Group 1	3:2:1	1.2 ng, 600 pg, 522 pg, 450 pg, 300 pg, 228 pg, 150 pg, 75 pg, 38 pg	2
3-person Group 1	10:5:1	3.2 ng, 1.6 ng, 1.4 ng, 1.2 ng, 800 pg, 608 pg, 400 pg, 200 pg, 100 pg	2
3-person Group 1	100:100:4	1.28 ng, 625 pg, 325 pg	2
3-person Group 2	1:1:1	1.2 ng, 600 pg, 300 pg, 150 pg, 75 pg, 38 pg	1
3-person Group 2	3:2:1	1.2 ng, 522 pg, 300 pg, 150 pg, 38 pg	2
3-person Group 2	10:5:1	3.2 ng, 1.4 ng, 800 pg, 400 pg, 100 pg	2
3-person Group 2	100:100:4	1.28 ng, 638 pg, 319 pg	2
4-person Group 1	4:3:2:1	2 ng, 1 ng, 870 pg, 750 pg, 500 pg, 380 pg, 250 pg, 125 pg, 63 pg	2
4-person Group 1	10:5:2:1	3.6 ng, 1.8 ng, 1.6 ng, 1.4 ng, 900 pg, 684 pg, 450 pg, 225 pg, 113 pg	2
4-person Group 1	100:100:100:6	1.28 ng, 625 pg, 325 pg	2
4-person Group 2	1:1:1:1	1.6 ng, 800 pg, 400 pg, 200 pg, 100 pg, 50 pg	1
4-person Group 2	4:3:2:1	2 ng, 870 pg, 500 pg, 250 pg, 63 pg	2
4-person Group 2	10:5:2:1	3.6 ng, 1.6 ng, 900 pg, 450 pg, 113 pg	2
4-person Group 2	100:100:100:6	1.28 ng, 638 pg, 319 pg	2

**Table 3 genes-13-02031-t003:** Propositions for simple LR product v. compound LR comparisons. Note that the designations “C1”, “C2”, etc. are generic and represent more than one combination of mixture contributors. For instance, under “Two true contributors”, C1 and C2 in proposition set (2) represent the three different combinations of two true contributors in a three-person mixture that could be paired together.

Compound LR Comparison Propositions (2 True Contributors)	
Simple LR product propositions	Compound LR propositions
$\frac{C1+\mathrm{Unk}}{\mathrm{Unk}+\mathrm{Unk}}*\;\frac{C2+\mathrm{Unk}}{\mathrm{Unk}+\mathrm{Unk}}$	$\frac{C1+C2}{\mathrm{Unk}+\mathrm{Unk}}$
$\frac{C1+\mathrm{Unk}+\mathrm{Unk}}{\mathrm{Unk}+\mathrm{Unk}+\mathrm{Unk}}*\frac{C2+\mathrm{Unk}+\mathrm{Unk}}{\mathrm{Unk}+\mathrm{Unk}+\mathrm{Unk}}$	$\frac{C1+C2+\mathrm{Unk}}{\mathrm{Unk}+\mathrm{Unk}+\mathrm{Unk}}$
$\frac{C1+\mathrm{Unk}+\mathrm{Unk}+\mathrm{Unk}}{\mathrm{Unk}+\mathrm{Unk}+\mathrm{Unk}+\mathrm{Unk}}*\frac{C2+\mathrm{Unk}+\mathrm{Unk}+\mathrm{Unk}}{\mathrm{Unk}+\mathrm{Unk}+\mathrm{Unk}+\mathrm{Unk}}$	$\frac{C1+C2+\mathrm{Unk}+\mathrm{Unk}}{\mathrm{Unk}+\mathrm{Unk}+\mathrm{Unk}+\mathrm{Unk}}$
**Compound LR comparison propositions (3 true contributors)**	
Simple LR product propositions	Compound LR propositions
$\frac{C1+\mathrm{Unk}+\mathrm{Unk}}{\mathrm{Unk}+\mathrm{Unk}+\mathrm{Unk}}$ $*\frac{C2+\mathrm{Unk}+\mathrm{Unk}}{\mathrm{Unk}+\mathrm{Unk}+\mathrm{Unk}}$ $*\frac{C3+\mathrm{Unk}+\mathrm{Unk}}{\mathrm{Unk}+\mathrm{Unk}+\mathrm{Unk}}$	$\frac{C1+C2+C3}{\mathrm{Unk}+\mathrm{Unk}+\mathrm{Unk}}$
$\frac{C1+\mathrm{Unk}+\mathrm{Unk}+\mathrm{Unk}}{\mathrm{Unk}+\mathrm{Unk}+\mathrm{Unk}+\mathrm{Unk}}$ $*\frac{C2+\mathrm{Unk}+\mathrm{Unk}+\mathrm{Unk}}{\mathrm{Unk}+\mathrm{Unk}+\mathrm{Unk}+\mathrm{Unk}}$ $*\frac{C3+\mathrm{Unk}+\mathrm{Unk}+\mathrm{Unk}}{\mathrm{Unk}+\mathrm{Unk}+\mathrm{Unk}+\mathrm{Unk}}$	$\frac{C1+C2+C3+\mathrm{Unk}}{\mathrm{Unk}+\mathrm{Unk}+\mathrm{Unk}+\mathrm{Unk}}$
**Compound LR comparison propositions (4 true contributors)**	
Simple LR product propositions	Compound LR propositions
$\frac{C1+\mathrm{Unk}+\mathrm{Unk}+\mathrm{Unk}}{\mathrm{Unk}+\mathrm{Unk}+\mathrm{Unk}+\mathrm{Unk}}$ $*\frac{C2+\mathrm{Unk}+\mathrm{Unk}+\mathrm{Unk}}{\mathrm{Unk}+\mathrm{Unk}+\mathrm{Unk}+\mathrm{Unk}}$ $*\frac{C3+\mathrm{Unk}+\mathrm{Unk}+\mathrm{Unk}}{\mathrm{Unk}+\mathrm{Unk}+\mathrm{Unk}+\mathrm{Unk}}$ $*\;\frac{C4+\mathrm{Unk}+\mathrm{Unk}+\mathrm{Unk}}{\mathrm{Unk}+\mathrm{Unk}+\mathrm{Unk}+\mathrm{Unk}}$	$\frac{C1+C2+C3+C4}{\mathrm{Unk}+\mathrm{Unk}+\mathrm{Unk}+\mathrm{Unk}}$

**Table 4 genes-13-02031-t004:** Propositions for unconditioned v. conditioned LR comparison. Note that this comparison applies to compound LRs as well as simple LRs; for instance, for the three-person mixtures, both the simple and 2-person compound LRs were compared to their conditioned counterparts.

Conditioned LR Comparison Propositions (1 Conditioned True Contributor)	
Unconditioned propositions	Conditioned propositions
$\frac{C1+\mathrm{Unk}}{\mathrm{Unk}+\mathrm{Unk}}$	$\frac{C1+C2}{\mathrm{Unk}+C2}$
$\frac{C1+\mathrm{Unk}+\mathrm{Unk}}{\mathrm{Unk}+\mathrm{Unk}+\mathrm{Unk}}$	$\frac{C1+C2+\mathrm{Unk}}{\mathrm{Unk}+C2+\mathrm{Unk}}$
$\frac{C1+C2+\mathrm{Unk}}{\mathrm{Unk}+\mathrm{Unk}+\mathrm{Unk}}$	$\frac{C1+C2+C3}{\mathrm{Unk}+\mathrm{Unk}+C3}$
$\frac{C1+\mathrm{Unk}+\mathrm{Unk}+\mathrm{Unk}}{\mathrm{Unk}+\mathrm{Unk}+\mathrm{Unk}+\mathrm{Unk}}$	$\frac{C1+C2+\mathrm{Unk}+\mathrm{Unk}}{\mathrm{Unk}+C2+\mathrm{Unk}+\mathrm{Unk}}$
$\frac{C1+C2+\mathrm{Unk}+\mathrm{Unk}}{\mathrm{Unk}+\mathrm{Unk}+\mathrm{Unk}+\mathrm{Unk}}$	$\frac{C1+C2+C3+\mathrm{Unk}}{\mathrm{Unk}+\mathrm{Unk}+C3+\mathrm{Unk}}$
$\frac{C1+C2+C3+\mathrm{Unk}}{\mathrm{Unk}+\mathrm{Unk}+\mathrm{Unk}+\mathrm{Unk}}$	$\frac{C1+C2+C3+C4}{\mathrm{Unk}+\mathrm{Unk}+\mathrm{Unk}+C4}$
**Conditioned LR comparison propositions (2 conditioned true contributors)**	
Unconditioned propositions	Conditioned propositions
$\frac{C1+\mathrm{Unk}+\mathrm{Unk}}{\mathrm{Unk}+\mathrm{Unk}+\mathrm{Unk}}$	$\frac{C1+C2+C3}{\mathrm{Unk}+C2+C3}$
$\frac{C1+\mathrm{Unk}+\mathrm{Unk}+\mathrm{Unk}}{\mathrm{Unk}+\mathrm{Unk}+\mathrm{Unk}+\mathrm{Unk}}$	$\frac{C1+C2+C3+\mathrm{Unk}}{\mathrm{Unk}+C2+C3+\mathrm{Unk}}$
$\frac{C1+C2+\mathrm{Unk}+\mathrm{Unk}}{\mathrm{Unk}+\mathrm{Unk}+\mathrm{Unk}+\mathrm{Unk}}$	$\frac{C1+C2+C3+C4}{\mathrm{Unk}+\mathrm{Unk}+C3+C4}$
**Conditioned LR comparison propositions (3 conditioned true contributors)**	
Unconditioned propositions	Conditioned propositions
$\frac{C1+\mathrm{Unk}+\mathrm{Unk}+\mathrm{Unk}}{\mathrm{Unk}+\mathrm{Unk}+\mathrm{Unk}+\mathrm{Unk}}$	$\frac{C1+C2+C3+C4}{\mathrm{Unk}+C2+C3+C4}$

**Table 5 genes-13-02031-t005:** Effect of increasing MCMC burn-in and post burn-in accepts on the false negative rate for 4-person compound LRs calculated for a 250 pg 4:3:2:1 mixture.

Fold Increase in Accepts	Burn-In or Post Burn-In	False Negative LRs in 10 Replicates	Average Decon Time
None	-	9/10	1.36 min
10x	Burn-in	10/10	2.30 min
10x	Post burn-in	4/10	10.71 min
20x	Burn-in	1/10	21.07 min
24x	Post burn-in	1/10	25.44 min
15x	Both	0/10	17.73 min
20x	Both	0/10	23.90 min
50x	Both	0/10	58.70 min

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
