# Peer review of "Compound and Conditioned Likelihood Ratio Behavior within a Probabilistic Genotyping Context"

_genes, 2022, doi:10.3390/genes13112031_

Round 1

Reviewer 1 Report

The manuscript describes a series of studies examining the relationship between simple LR products and compound LRs with and without conditioning.  To date, this is the most comprehensive study examining this relationship and provides important information to the analysts that use probabilistic genotyping.

General Comment:

-          To easily counter some of the arguments and publications that state there is a lack of data supporting various aspects of probgen, it would be beneficial to have a brief summary of the number of profiles examined and the total number of LRs calculated for this study somewhere in the manuscript (Conclusions section maybe).

Materials and Methods:

-          Lines 138 – 147: Would it be worth mentioning the stutter that was modeled since this can vary between laboratories? 

-          Lines 145 – 146: Suggest adding “per chain” to the burnin and readout list (e.g. 10,000 accepts per chain…)

-          Line146 (and throughout manuscript): This is probably a matter of preference, burnin vs burn-in

-          Line 146 (and throughout manuscript): I am not familiar with the term “readout” vs “post-burnin accepts”

Results:

-          Somewhere in the results section (or maybe the Materials and Methods) the authors should state that exclusions are capped at a log LR of -30.  I apologize if it is stated and I missed it.

-          Suggestion: When discussing the false negatives (Section 3.3), it may be helpful to list the % of false negative results for each category in Figure 3.  Due to the overlapping circle data points, it becomes difficult to see just how many false negatives there are for the 3 and 4 combined. 

-          Section 3.7: The authors state that “a very low level contributor produced unexpectedly high peak data at one locus…” The egram is provided in the supplementary, however, it would be more informative if the authors could give an estimated expected peak height for the alleles in question so the reader can know what the magnitude of the differences are that caused the problem.

Author Response

Please see the attached Word document for our responses to your review of our publication.

Reviewer 2 Report

Review comments for “Compound and conditioned likelihood ratio behavior within a probabilistic genotyping context”.  This paper addresses an interesting aspect of LR behavior that, to my knowledge, is not well known and therefore useful to have quantitative data on.  The paper is well written, organized, and succinct.  My comments are mostly points of clarification that are suggested to ease the readers’ ability to fully understand the work done.  The authors clearly performed a lot of work, but the depth of the data is sometimes lost in the presentation of the paper by not including details of the particulars of the experiments and examples of specific data values.

Section 1.

Line 47 – 50 The statement “Generally, when the simple LRs for multiple individuals are greater than 1, conditioning on one or more individuals increases the magnitude of the LR for the remaining individuals, if all individuals are true contributors to the mixture.” is certainly intuitive, however, this statement should be supported by a reference.

Line 57 In the sentence “On the other hand, the LR best representing the evidence may include all questioned individuals” the word “best” feels like a value judgement, as the inclusionary proposition can change, depending on the circumstances of the case.  Perhaps consider replacing “best” with something like “the LR most thoroughly representing” or “the LR most appropriately representing”.

Line 99 -100 While I completely agree with the statement “Such information can assist the casework analyst in evaluating the statistical weights of the LRs calculated in a case at hand.” I think the sentence would benefit from an example(s) of how the information can assist the analyst.  As is, the sentence is a nice lead into the benefits of the study, but as a reader I would know like some specific instances of how it will help me in my data interpretations.

Section 2.

Lines 106 and 512 The authors specify that samples were collected with “informed consent” but provide no further information.  Was there an IRB, or equivalent, that they had approval from?  Was the informed consent documented?

Line 120 – 122 The authors describe how they assembled their mixture combinations based on quantitation data.  Was any attempt made to confirm the target mixture ratios using the peak heights in the electropherograms to backcheck that the quantitation data was accurate and gave the allele peak data needed to proceed to probabilistic genotyping with confidence?

Line 129 – 131 The authors specify that some mixtures were amplified “either singly or in duplicate”.  No explanation is given for why there was a difference in the number of replicate amplifications.  Why was a standard number of amplifications not used for each mixture sample?  It is also not made clear in the paper how data from the mixtures with two replicate amplifications was used.  Was one of the replicates chosen for analysis with STRmix?  If so, by what criteria was that one chosen?  Were data from both replicates used and averaged?  This is my most serious concern with the paper, as it is just not clear what data were used for the STRmix analyses, and a more thorough explanation needs to be included.

Table 2 The information in Table 2 is clear and relatively easy to understand, however the actual amounts of individual contributor template DNA are not listed anywhere.  In some mixtures it is easily intuited by the reader (1:1, 1:1:1, etc.) but in many others it is not, nor should the reader really have to do the math.  This is especially relevant as later in the paper (Sections 3 & 4) template DNA features strongly in explaining the observed results.  While some of those results are for the LRs from the combined propositions, it would still be useful to the reader to understand the amounts of template for the contributors to the mixtures.

Table 3 A minor point in Table 3 is the use of the symbol (+) to indicate the word “and”.  This can be a bit confusing as it outwardly implies an additive calculation function within the LR equation.  The authors may want to consider changing to the “&” symbol.

Section 3.

Line 248 The sentence “compound LRs largely holds over the entire DNA input range for 2- and 3-person” would benefit from explicitly stating what the DNA input range is in pg, rather than making the readers go back to the tables and figure it out.

Line 264 – 266 The sentence “The 2-person mixtures are so well resolved that only the 1:1 mixtures show any increase in the compound LR compared to the simple LR product” would also benefit by adding information about which of the template DNA quantities are being referred to here.  All of them? Just a subset?  Figure 1a shows what appears to be four of the 1:1 data points, but no information on corresponding input values.

Figure 3b lists a mixture ratio of 100:100:4 for the 4-person mixtures in panel b which is not listed in Table 2 and is not in panels a or c of the figure.  Panel c lists a ratio of 100:100:100:4, which is also not in Table 2.  Are these typos?

Lines 305 – 311 Another instance as described above, the authors discuss the importance of DNA mixture input in false negative LRs, but do not specify what those input values are, leaving the reader to infer values.  I have no doubt that the statements in the specified lines are correct, it would just really help to have more specific data to fully appreciate the points being made.  It would also answer questions such as, were all mixture contributors in low-template quantities, all contributors except majors in low-template, or just the minors in low-template?

Line 322 Another minor point, but the lines preceding 322 refer to just Supplemental Figure A2a and the readers attention should be focused on just that panel of the figure, rather than A2b at this point.

Line 338 – 340 The sentence “If the complexity ceiling is the issue causing the false negative LRs and large differences between the point estimate LR and HPD LR, then an increase in MCMC accepts would likely be an appropriate solution.” should have a reference.

Lines 348 – 350 It would be interesting to know which (all?) contributors to the 250pg 4:3:2:1 4-person mixture had the false negative LR(s) and what their input DNA values were.

Lines 366 – 369 The authors state that “No false negative LRs were observed” for the 4-person mixtures re-run with 20x MCMC accepts, however in Figure 4 panels a and b, there appear to be data points from the 100:100:100:6 ratio mixtures that fall into the negative ends of both axes.  Is this correct? Some mention of these data points would be useful, even if it is just the case that both analysis conditions resulted in false negative LRs.  This comment holds for similar data points observed in figures 5 & 6.

Figure 4 panels b and c have the same mixture ratios (100:100:4) and (100:100:100:4) as in Figure 3.  Are these typos?

Section 4.

Lines 436 – 439 The sentence “Furthermore, the typical amount of increase was dependent on both the DNA input level as well as the mixture composition; specifically, higher DNA inputs and more ambiguous genotype combinations led to greater increases in the compound LR relative to the simple LR product.” would benefit from some specific examples of values from the data to demonstrate the point.

Section 5.

Lines 491 – 494 The sentence “While not all of the compound LRs were greater than their simple LR product equivalents, a vast majority were equal or greater, and in a predictable way that depended primarily on mixture proportions.” is another example of the need for more depth.  The study has a lot of very interesting data and specifying the mixture proportions referred to in this sentence would help the reader to fully grasp the point being made.

Author Response

(The authors gave the same response as above.)

Round 2

Reviewer 2 Report

The revisions made by the authors have been thorough and addressed my comments well.

I did notice the following on review;

Supplementary Figure A1a & b in the first version (red line) uses "Simple" to denote the data points in blue. In the revised manuscript (black line) this has been changed to "Single".

Supplementary Figure A2a uses "Simple" to denote the data points in blue, but A2b uses "Single" for the data points in blue in both the original and revised manuscripts.

I just wanted to confirm that the terms "Simple" and "Single" were intentional in their uses in each panel and not inadvertent.

Author Response

Please see the attached Word file for our response to your review of our publication.
